# A Review on the Feedstocks for the Sustainable Production of Bioactive Compounds in Biorefineries

**Sebastián Serna-Loaiza \*** , **Angela Miltner** , **Martin Miltner** **and Anton Friedl**

Institute of Chemical, Environmental and Bioscience Engineering, TU Wien, 1060 Vienna, Austria;
angela.miltner@tuwien.ac.at (A.M.); martin.miltner@tuwien.ac.at (M.M.); anton.friedl@tuwien.ac.at (A.F.)
**\*** Correspondence: sebastian.serna@tuwien.ac.at; Tel.: +43-1-58801-166262

**Abstract:** Since 2015, the sustainable development goals of the United Nations established a route map to achieve a sustainable society, pushing the industry to aim for sustainable processes. Biorefineries have been studied as the technological scheme to process integrally renewable resources. The so-called "bioactive" compounds (BACs) have been of high interest, given their high added value and potential application in pharmaceutics and health, among others. However, there are still elements to be addressed to consider them as economic drivers of sustainable processes. First, BACs can be produced from many sources and it is important to identify feedstocks for this purpose. Second, a sustainable production process should also consider valorizing the remaining components. Finally, feedstock availability plays an important role in affecting the process scale, logistics, and feasibility. This work consists of a review on the feedstocks for the sustainable production of BACs in biorefineries, covering the type of BAC, composition, and availability. Some example biorefineries are proposed using wheat straw, hemp and grapevine shoots. As a main conclusion, multiple raw materials have the potential to obtain BACs that can become economic drivers of biorefineries. This is an interesting outlook, as the integral use of the feedstocks may not only allow obtaining different types of BACs, but also other fiber products and energy for the process self-supply.

**Keywords:** bioactive compounds; biomass; bio-based economy; circular economy; sustainability

---

## 1. Introduction

The need to change the economy and society in general from oil-based to bio-based systems is a topic that has been in discussion for decades [1,2]. In addition, sustainability—for which multiple definitions have been proposed—is a topic that has been in discussion since the 1980s. However, in principle, the various definitions relate to the definition of the World Commission on Environment and Development (WCED), which states that sustainable development is the "development that meets the needs of the present without compromising the ability of future generations to meet their own needs" [3]. In recent decades, multiple frameworks, objectives, and goals were proposed to address this topic. Since 2015, the sustainable development goals (SDG) of the United Nations established a route map in order to achieve a more sustainable society, aiming to reach these goals by 2030 [1]. This framework involves 17 goals and intends to associate the three constituent parts (ecology, economics, and society) through 169 targets that aim to achieve sustainable development.

This framework pushes the industry to aim for sustainable production processes, suggesting that processes must focus on ecological, economic, and social factors, which applies both for processes already in operation, as well as in the design stage of new ones. Reducing the dependency on fossil-based energy and materials, using raw materials more intensively, improving energy efficiency, using renewable raw materials and biomass, among many others, are some of the approaches that have been proposed for the industry in order to achieve sustainable production. Within these approaches,

biorefineries have been studied as a technological scheme to process integrally renewable resources and add value to biomass [4]. A biorefinery is the adaptation of a petroleum refinery scheme (one feedstock—raw oil—to produce multiple products) to a biomass-based scheme. Most definitions state that a biorefinery is a facility that integrates biomass conversion processes and equipment to obtain multiple products from biomass. Industrial biorefineries have been identified as the most promising route for the establishment of a bio-based industry [5,6].

A great variety of feedstocks (edible crops, residual biomass, algae, and non-edible crops), technologies (biochemical, chemical, thermochemical, catalytic, and physical), and products have been researched in recent years [7]. The proposed products are biomolecules and natural chemicals, biomaterials, biofuels, bioenergy, and food/feed. This broad range is key, given that a shift into a bioeconomy requires the production of both energy and products used in different industrial and daily life applications [4,8]. Among these, the so-called "bioactive" compounds have been of high interest, given their high added value and their potential application in pharmaceutics, health, nutrition, and cosmetics, among other fields [9]. However, their definition is still ambiguous and unclear. In broad terms, they can be classified as compounds that are able to interact with one or more components of living tissue by presenting a wide range of probable effects [10].

However, despite the applicability and economical interest in these compounds, there are still certain elements that need to be analyzed in order to consider them as economic drivers of sustainable production processes. First, bioactive compounds (BACs) can be obtained from many sources, either by extraction from their natural matrix or by producing them via fermentation and synthesis, among others. Therefore, it is important to identify raw materials that can be used for this purpose. Second, for the production process to be sustainable, the extraction/production of the bioactive compound itself should not be considered alone but also the valorization of the remaining components of the raw material. Hence, identifying the components of the matrix of the feedstock becomes of great importance, as this determines the possible products to be obtained. Finally, the availability of the feedstock plays an important role, because it directly affects the scale of the process and this is closely related to transport and logistics.

Based on the aforementioned elements, this work consists of a review on the feedstocks for the sustainable production of bioactive compounds in biorefineries, covering the different mentioned items (type of bioactive compound, composition, and availability, among others). Then, some example biorefineries are proposed using different types of raw materials (hemp, grape vine shoots, and wheat straw) and the respective by-products that could be obtained are also mentioned. Section 2 identifies some key elements to be considered for the sustainability of a biorefinery (technical, economic, environmental, and social) and establishes that the focus of this work is on the intensive use of the feedstock. Section 3 summarizes the different concepts of "bioactive compounds", the different types of BACs, the sources from which they can be obtained and some examples of sources, with their respective BACs and reported applications. Then, Section 4 describes the different components that make up a feedstock, in addition to the BAC, and presents different technologies used to valorize these remaining fractions. Finally, Section 5 selects different potential feedstocks for the biorefineries to produce BACs, based on different criteria, and three study cases are described, which put together the information available in the previous sections.

As a main conclusion, it was observed that multiple raw materials that are currently used have potential in obtaining BACs that can become the economic drivers of the biorefinery. In addition, the further valorization of the feedstocks allows broadening the range of products (including other BACs as antibiotics), and, ultimately, the remaining solids can be still used for the production of energy.

## 2. Important Aspects to Consider for the Sustainability of a Biorefinery

Before covering specific details about the feedstocks for the sustainable production of BACs in biorefineries, it is necessary to address the topic of "sustainability", and how a process can be made sustainable. In the broadest sense, sustainable development is a principle that brings together human

development and ensures that nature—including the natural resources and ecosystem upon which the economy and society depend—is sustained. In terms of industry, this means that both processes already in operation, as well as in the design stage of new ones must focus on ecological, economic, and social factors [1]. In addition, the underlying technical aspects keep playing an important role especially in terms of the economic and ecological performance of a process.

It is important to bear in mind that the topics of sustainability and sustainable development are broad enough and are still under discussion that could be the topic itself of a review paper. Moreover, in this same sense, they are key driving topics that cannot be simply overpassed. Due to this, the focus of this section, rather than deepening in the broad definitions and approaches of sustainability, it will be mentioning some key bullet points that must be considered for the design of a sustainable biorefinery. These key points represent the most immediate aspects that can be addressed since the design stage of a biorefinery. Therefore, they should be considered as a first level of decision in process design [11].

Figure 1 shows some important points to be considered in the design of sustainable biorefineries. From the economy side, the profitability of the process is an obvious key aspect, which implicates finding economic drivers for biorefineries to achieve a positive net present value or income margin during the lifetime. From the ecology side, the assessment of the lifecycle of the biorefinery will show the environmental impacts; these should be covered from a broader perspective that includes not only the global warming potential, but also other critical aspects as the pressure on the land and water use [12]. Finally, from the social side, the first direct social impact can be job generation and a second can be the offer of renewable/sustainable options to consumer in order to decrease the dependency on the fossil-based economy. Although the social sphere can be the furthest aspect to be influenced/addressed since the design stage of a biorefinery [13], job generation and the offer of alternatives are the most inherent elements that can be related to the design. However, in recent years, multiple methodologies and indicators have been developed to perform social life cycle assessment of processes in order to decrease the gap between technical and social aspects of processes [13–15].

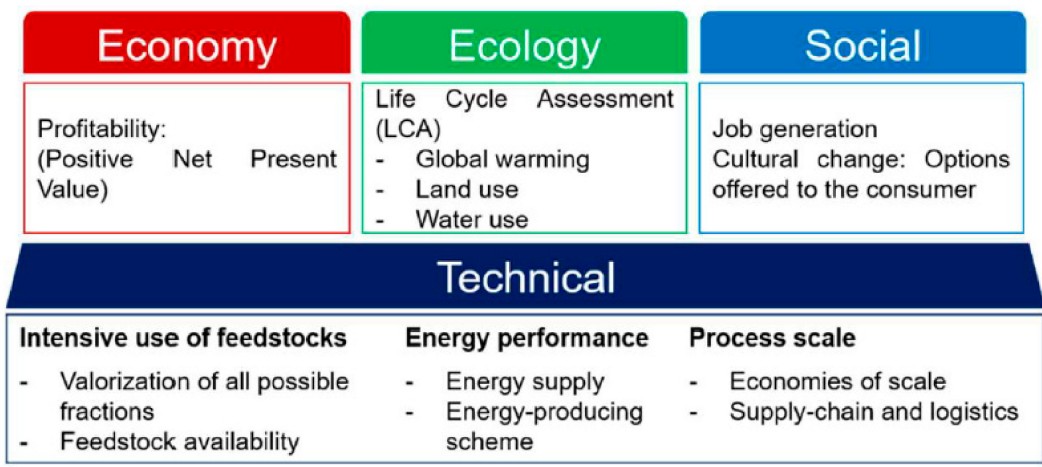

**Figure 1.** Important points to be considered in the design of sustainable biorefineries.

Beneath the three previous elements underlies the technical assessment as the base tool for the design, which requires the consideration of important aspects that can have a direct impact of the process sustainability such as an intensive use of the feedstock, the energy performance, and the process scale. These elements play very important roles that will be reflected in the economy, ecology, and social spheres.

Regarding the *intensive use of feedstocks* and *process scale*, biomass and renewable feedstocks are made of different components, each with possible applications that should be valorized. The recent state of the art and research have shown a wide range of applications for a feedstock, which indicates

different applications, processes, and products. This topic will be developed in more detail in Section 4 of this review. The availability of the feedstock plays an important role, because it directly affects the scale of the process. Small- and large-scale processes have different economic behavior (economies of scale), and this is closely related to the supply-chain and logistics required for a given scale [4]. Therefore, feedstock availability represent a crucial point in the technical feasibility of a biorefinery.

Regarding *energy performance*, biorefineries should consider the use of remaining solids streams for the production of energy, which could supply partially or totally the energy requirements of the process. There are different thermochemical and biochemical technologies (gasification, anaerobic digestion, pyrolysis, and combustion, among others) that could be used for this purpose. However, each one of them may be feasible under given scales. This is especially important for biorefineries focused on obtaining chemicals, materials, and products, considering that after the subsequent processing of the feedstock there will be solids fractions that might be able to produce energy. This topic is discussed further in Section 4.2.

In this specific review, the focus will be on the *intensive use of feedstocks*, as this is the first connection point between the production of bioactive compounds, their sources, the composition of the feedstock, and the further possible valorization that can be done under a biorefinery scheme.

## 3. Bioactive Compounds and Raw Materials

This first section addresses briefly some generalities about bioactive compounds, the different definitions, types, sources, and the current technologies for their production.

### 3.1. Definition of "Bioactive"

As mentioned before, research regarding BACs has increased significantly and the wide variety of compounds and origins/sources has generated a broad number of definitions. The two most common application fields for BACs are pharmaceutical and food supplementation. For this reason, this first section will briefly cover some of the definitions and the context for which they are given.

Bioactive compounds are substances that have a biological activity and generate a response in living tissues. In addition, this response must generate beneficial effects on health [10]. According to the National Cancer Institute of the United States, these are chemicals found in small amounts in plants and certain foods (such as fruits, vegetables, nuts, oils, and whole grains), which have actions in the body that may promote good health [16]. Regarding food applications, some authors specify that these are non-essential nutrients that are extensively studied to evaluate their effects on health [17].

Another aspect to be considered in the definition of BACs is the origin. Generally, despite the definition of BACs considers a biological origin, this does not mean that a synthetic molecule cannot have a biological activity. For this reason, some authors have included in their definition the BACs derived from nature and synthetic products. Multiple cosmetics, medicine, and agricultural products, among others, have bioactive principles based on synthetic molecules (e.g., sulfamethoxazole, lomefloxacin, and linezolid, among others) [18]. In this same sense, other definitions of bioactive compounds consider the origin not only of those compounds extracted from a natural matrix, but also those produced via fermentation or via synthesis. Coumarin can be an example of both a natural and a synthetic BAC, as it can be extracted from a natural matrix or produced via synthesis through the Perkin reaction [19]. Regarding the compounds obtained via synthesis, it is important to mention that these are compounds that were initially identified from a natural source, but that an alternative chemical synthesis method has been developed for their production, instead of an extraction from the natural matrix.

In strict terms, the biological activity of a given chemical can have beneficial or harmful effects. This depends on factors such as the chemical structure, dosing, and application conditions, etc. The most typical example of this behavior can be the Botulinum toxin (BTX, botox). Apart from the cosmetic applications, this compound is used to treat a number of disorders characterized by overactive muscle movement (post-stroke spasticity, post-spinal cord injury spasticity, and spasms, among others) [20].

However, this compound can be a lethal toxin with $LD_{50}$ values of 1.3–2.1 ng/kg. Hence, BACs can also have biocide applications, which have a positive effect on a living organism by fighting against another. This would be the case of antibiotics [21]. Therefore, BACs are beneficial for a specific target organism under given application conditions.

Now, to summarize the different definitions, bioactive compounds are secondary metabolites that have the ability to interact with components of living tissue, presenting a wide range of probable effects aimed to have beneficial effect on the target organism. Its origin can be natural (plant, animal, or microorganism based) or synthetic.

### 3.2. Types and Sources of BAC

The following aspects to be addressed are the types of bioactive compounds and the different sources from which they can be obtained. These two elements are related, given that the source determines the types of compounds that can be obtained. For example, when referring to plant-based sources, BACs generally refer to phenolic compounds; however, for those produced via fermentation, they generally refer to antimicrobial substances active against bacteria or some specific compounds with pharmaceutical applications.

#### 3.2.1. Types

Phenolic Compounds

One of the most studied group of compounds due to their bioactivity are phenolic compounds. These are secondary metabolites of natural sources, which play roles associated to the growth, reproduction, protection, and sensory characteristics [22]. These are classified in phenolic acids, flavonoids, stilbenes, coumarins, lignins, and tannins.

*Phenolic acids* act as a building material by forming bridges with cellulose, hemicellulose, and pectin. These are divided into hydroxycinnamic acids (p-coumaric, caffeic, ferulic, and sinapic acids) and hydroxybenzoic acids (p-hydroxybenzoic, protocatechuic, vanillic, syringic, and gallic acids). These compounds have been attributed antitumor, antimicrobial, and antioxidant properties [23].

*Flavonoids* have a three-ring structure with the form C6–C3–C6. These are divided into flavones, flavonols, flavanones, flavanonols, flavanols, isoflavones, and anthocyanidins. These groups have been attributed anticancer, anti-inflammation, and antivirus properties, and the ability of reducing cardiovascular diseases and type 2 diabetes [24,25].

*Stilbenes* have a C6–C2–C6 carbon skeleton. They have antibacterial, anti-inflammation, and antitumorigenesis bioactivities, and cardioprotective effects, among others. Some stilebenes such as kobophenol-A/-B and resveratrol have been tested as antibacterial against *Staphylococcus aureus* and inhibition of Parkinson's and Alzheimer's diseases, respectively [26,27].

*Coumarins* have a C6–C3 skeleton and an oxygen heterocycle. They are found both in plants and microorganisms (*Streptomyces* and *Aspergillus* species) [19]. They show a wide range of bioactive applications such as anti-inflammatory, antimicrobial, antiviral, antioxidant, antiasthmatic, and anti-Alzheimer's [28,29].

*Tannins* are water-soluble phenolic compounds with the ability of precipitating alkaloids and proteins. Due to the presence of phenolic rings in their structure, they can act as electron scavenger, hence having a significant antioxidant potential [30].

The final group of phenolic compounds is *lignins*, which are a group of hetero-polymers formed by p-coumaryl alcohol, coniferyl alcohol, and sinapyl alcohol. Lignin is one of the most abundant organic polymers on earth, and most of its applications have been directed to the chemical industry, materials and energy. However, in recent years some research has been done for the extraction and production on micro- and nano-lignin, especially focused on its bactericidal, UV-blocking, and antioxidant properties [31].

Antibiotics

After covering the first group of BACs, the second group to be addressed is antibiotics. These compounds are of high interest for society as more bacteria continue to develop resistance to currently produced antibiotics. This has led to an intense research in topics such as large-scale production and the screening of antibacterials against a wide range of bacteria. Antibiotic production can be grouped into three methods: Natural fermentation, semi-synthetic, and synthetic. This will be briefly discussed in the following section.

Now, regarding the strains used for the production of antibiotics, first antibiotics naturally produced came from fungi as the *Penicillium* and soil bacteria that produced streptomycin and tetracycline. Semisynthetic production of antibiotics has currently the highest share in the production of antibiotics. This has been achieved by modifications of the wild microorganisms so that the production is enhanced. Some techniques as mutation (ultraviolet radiation, x-rays, or certain chemicals), selective reproduction, and metabolic engineering have been used for this purpose. Gene modification techniques have gained significant strength in this particular field as it allows combining natural features of certain microorganisms and fungi with the insertion of genes coding for the metabolism of given antibiotics [32]. Research in this aspect has opened the field going from bioinformatics—used for the identification of specific gene clusters that can metabolize compounds [33]—to genetic modification and inclusion of new metabolic features on microorganisms—for the production of antibiotics and up to the testing of new compounds for their biocide activity [34]. In addition, in order to overcome the barriers (e.g., structural complexity) posed by the chemical modification of natural products derived from fermentation, research has also focused on developing complete synthesis routes for the production of antibiotics [35].

Other Possible Bioactive Compounds

In addition to the phenolic compounds and the antibiotics, there are certain compounds outside of these categories, which are found in specific plants/fruits, and that have potential as BACs. Some examples of these compounds are carotenoids, curcuminoids, and cannabinoids, among others.

*Carotenoids* are pigments (yellow, orange, and red) produced by plants, algae, bacteria, and fungi. Pumpkins, carrots, corn, and tomatoes have their characteristic color because of the presence of carotenoids. Their biological function is to absorb light for the photosynthesis and to protect chlorophyll from degradation. For this reason, they have been considered to have antioxidant function. There have been studies about their beneficial effects as protection against cancer (lung, head, prostate, breast) [36,37].

*Curcuminoids* are natural phenols that produce yellow color. Curcumin is the most representative molecule of this group, but due to its low solubility, different derivatives have been developed. Curcumin and some derivatives as desmethoxycurcumin, bisabolocurcumin, and bisdemethoxycurcumin have been tested for their antioxidant activities [38].

*Cannabinoids* are a series of chemical compounds that act on the cannabinoid receptors (endocannabinoid system), altering the neurotransmitters of the brain. Despite some of these compounds are being produced synthetically, the great majority are phytocannabinoids that are found in *Cannabis* spp. plants. The best-known cannabinoid is tetrahydrocannabinol (THC), which has psychoactive effects. This is the main reason why cannabis plants are considered illegal and are controlled in several countries. However, THC is only one of at least 100 different cannabinoids that can be extracted from cannabis and have been studied for pharmaceutical and nutritional applications [39]. For example, cannabigerol (CBG) has been studied for treating sclerosis and glaucoma; cannabidiol (CBD) to treat schizophrenia and epilepsy; and cannabidivarin (CBDV) to treat epilepsy, among others [40]. Among the cannabis plants, *C. sativa*, also known as hemp or industrial hemp, has been identified for having high contents of CBD without significant content of THC. For this reason, the use of this plant has increased considerably over the last decade [41].

### 3.2.2. Sources

After covering some of the most important aspects about the current BACs and their applications, the next step to cover is the possible sources for their production. As discussed in the previous section, BACs are secondary metabolites of different kingdoms (plants, fungi, or even monera). For this reason, this section will address the sources (plant- and microorganism-based), the type of BACs in these sources and applications.

Table 1 shows different examples of sources where BACs could be obtained. Fruits, legumes, and seeds are potential sources of compounds that could be extracted and purified for further applications. In addition, not only these parts of the plants have BACs, but also other unused fractions resulting from the processing of the same (seeds, vine shoots, peels, pomace, spent pulps, etc.). This context offers the possibility of further consideration of using fractions that are considered as residues, having an integral use of the feedstocks and decreasing the generation of residues.

**Table 1.** Sources of different bioactive compounds (plant-based), type of bioactive compound, and applications. Adapted from (Shahidi and Yeo, 2018) [24].

| | Source | Type | Bioactivity/Application | Ref. |
|---|---|---|---|---|
| **Phenolic compounds, Carotenoids, Curcuminoids, Cannabinoids** | Hemp | Cannabinoids | Relieves convulsion, inflammation, anxiety, and nausea | [40] |
| | Wheat straw | Nano-lignin | Bactericidal; Antioxidant | [31] |
| | Grape (vine shoots, seed) | Resveratrol Kaempferol Quercetin | Inhibition of lipase activity; Reduced risk of cardiovascular disease; Bactericidal | [42,43] |
| | Apple | Phenolic acids Flavonols Dihydrochalcones | Reduced risk of cardiovascular disease; Decreased cholesterol level; Reduced risk of type 2 diabetes | [44] |
| | Citrus (orange, lemon, grapefruit) peel | Flavonoids | Antitumor; Antiatherosclerosis; Antibacterial; Reduces blood cholesterol | [9,45,46] |
| | Black rice | Flavones Tannin Anthocyanidins | Antiatherosclerosis; Antitumor; Antiallergic | [24] |
| | Blackberry | Phenolic acids Flavonoids Tannins | Anticancer | [47] |
| | Apricot | Coumarins | Antioxidant; Antimicrobial | [28,29] |
| | Soybean | Anthocyanins | Anti-inflammation | [48] |
| | Aloe vera | Catechin Quercetin | Anticancer; Anti-inflammation | [49] |
| | Coffee | Gallic acid Chlorogenic acid | Antioxidant | [50] |
| | *Alpinia officinarum* | Phenolic acids Flavonols | Anticancer; Anti-inflammation | [51] |
| | Olive | Flavonoids | Bactericidal; Antioxidant | [52] |
| | Tomato | Lycopene | Antioxidant; Anticancer | [53] |
| | Curcuma | Curcumin Bisabolocurcumin | Anti-inflammation | [54] |

Table 2 shows different examples of sources where BACs could be produced (antibiotics). From microorganisms producing antibiotics, there are two elements to be highlighted. First, there is a wide variety of substrates, which these microorganisms can use. This is a very important feature, given that both first and second-generation feedstocks can be used to produce pentoses, hexoses, and even other compounds as starch and glycerol can also be used as substrate. Second, with the current advances of metabolic engineering, well-known microorganisms are being studied in order to develop a heterologous expression of proteins that could synthesize antibiotics and overcome drawbacks as the inhibition by certain compounds, broaden the range of carbon sources or even potentiate the production [55].

**Table 2.** Sources of different bioactive compounds (microorganism-based), used substrate, and applications.

| | Microorganism | Antibiotic | Substrate | Bioactivity/Application | Ref. |
|---|---|---|---|---|---|
| **Antibiotics** | *Streptomyces hygroscopicus* | Geldanamycin | Pentoses, Hexoses, Glycerol, Starch | Antitumor; Inhibits nuclear hormone receptors | [56,57] |
| | *Acremonium chrysogenum* | Cephalosporin | Glycerol, Hexoses | Skin and soft tissue infection | [58] |
| | *Saccharopolyspora erythraea* | Erythromycin | Starch, Hexoses | Respiratory tract infections; Skin infections; Chlamydia; Pelvic inflammatory; Syphilis | [59] |
| | *Streptomyces griseus* | Streptomycin | Starch, Rice bran, Sucrose | Tuberculosis; Endocarditis; Brucellosis | [60] |
| | *Streptomyces aureofaciens* | Tetracycline | Hexoses | Acne; Cholera; Brucellosis; Malaria; Syphilis | [60] |
| | *Streptomyces coelicolor* | Undecylprodigiosin | Pentoses | Antimalarial activity | [61] |
| | *Amycolatopsis orientalis* | Vancomycin | Starch, Dextrin | Skin infections; Bloodstream infections; Endocarditis; Meningitis | [62,63] |

*3.3. Technologies*

After covering the types and sources of BACs, the next stage consists of briefly describing the current technologies for their extraction/production. Extraction techniques refer to plant-based sources that contain BACs, while production techniques refer to microorganism-based production of BACs.

3.3.1. Extraction Techniques

Conventional Extraction Techniques

Before describing the conventional extraction techniques, it is necessary to talk first about solvents. The conventional methods depend mainly on the chosen solvent, as this will influence the molecular affinity, mass transfer, cost, and environmental safety. Some of the most used solvents are mentioned below. Water is used for the extraction of anthocyanins, tannins, saponins, and terpenoids; ethanol is used for tannins, polyphenols, flavonols, terpenoids, alkaloids, and cannabinoids; methanol, for anthocyanins, terpenoids, saponins, tannins, flavones, and polyphenols; chloroform, for terpenoids and flavonoids; ether, for alkaloids and terpenoids; and acetone, for flavonoids [64–66]. Other relevant

variables for conventional extraction techniques are the temperature, solid/solvent ratio, extraction time, and number of extractions. These variables strongly influence the mass transfer. For example, higher temperatures, longer extraction times, and a higher number of extractions will increase the specific time at which the solvent is in contact with the solid, which facilitates the transfer of the extractives from the solid matrix to the solvent. On the other hand, having higher amounts of solid for a given amount of solvent does not implicate a higher extraction, as the solvent will not be able to reach effectively the whole mass of the solid; for this reason, the solid/solvent ratio plays an important role.

Now, covering the extraction techniques, *Soxhlet extraction* is one of the most used methods, even as a comparison standard for new techniques. It consists of a thimble where the dry sample is placed, which is located in the superior section of the equipment. On the lower section, the chosen solvent is placed and heated until evaporation. Then, a condenser brings the liquid solvent to the section with the sample, until it reaches a given height, where a siphon aspirates the solvent to the evaporation vessel. *Maceration* is another technique, in which a milled solid is mixed with the chosen solvent. After this, the solid–liquid mixture is pressed and filtered to separate a liquid fraction rich in the extracted compounds. *Hydrodistillation* is another conventional method for extraction of BACs and essential oils. It uses water/steam as extractant and the solid is packed inside a column. Then, the extractant is evaporated and condensed, as it happens in a normal distillation process, and a fraction of the condensed liquid is removed, which is enriched on the BACs. *Standard solid–liquid extraction* is another conventional extraction method, which consists of a fixed bed plant material put in contact with a warm/cold liquid at atmospheric conditions [67].

Despite being the most used methods for extraction, conventional techniques have some drawbacks as high extraction times, low selectivity, thermal decomposition of thermolabile compounds, requirement of pure solvents and in some cases, the performance is strictly related to the recovery of the solvent [67]. Therefore, some non-conventional techniques have been proposed to overcome these drawbacks. These new techniques can be divided into two groups: Those focused on increasing the accessibility of the feedstock to enhance the performance of a subsequent conventional technique, and those focused on extraction itself. These will be covered in the next sections.

Non-Conventional Extraction Techniques: Conventional-Technique Enhancers

As previously mentioned, these techniques are focused on enhancing conventional techniques. The application of these aims to decrease variables such as the extraction time, energy and use of solvent, and increasing variables such as the mass transfer and extraction efficiency.

*Ultrasound-assisted extraction (UAE)*: This technique uses ultrasound waves (around 20 kHz to 100 MHz) that pass through a medium while creating compression and expansion, which produces cavitation (production, growth, and collapse of bubbles). This cavitation process intensifies mass transfer and accelerates the access of a solvent to cell materials. The most important variables are the temperature, pressure, frequency, and time of sonication [68].

*Pulsed-electric field extraction (PEF)*: This technique applies a pulse-electric field to a solvent–solid mixture, through which the molecules of the solid separate creating pores. The application of the electric fields induces an electroporation of the cell, which permeabilizes the membrane, improving the mass transfer of the intracellular components [69]. These pores increase the accessibility and the surface area, hence improving the mass transfer and the access of the solvent. The most important variables are the field strength, specific energy input, pulse number, and temperature [70].

*Enzyme-assisted extraction (EAE)*: Some BACs are located within the cells while others are part of the lignocellulosic matrix. For this reason, enzyme-assisted extraction has been proposed in order to release compounds bound to this complex. Different enzymes (cellulases, α-amylases, and pectinases) are used to hydrolyze the cell wall. The most important variables are the enzyme concentration, particle size of plant materials, solid to water ratio, and hydrolysis time [71].

*Microwave-assisted extraction (MAE)*: Microwaves are waves in the electromagnetic field ranging from 300 MHz to 300 GHz, which are submitted to a given medium and change the dipole configuration.

By doing this, ions continuously flow in different directions inside the medium, heating through conduction and separating polar components. The most important variables are the frequency, temperature, pressure, and operation time [72].

Non-Conventional Extraction Techniques: Extraction Techniques

The techniques in this category are focused on the extraction itself and are able to provide conditions to extract bioactive compounds from their matrix.

*Pressurized liquid extraction (PLE)*: This technique extracts the compounds of interest by applying high pressure, through which the solvent remains liquid. This high-pressure condition increases the solubility and mass transfer, and decreases the viscosity and surface tension. The most important variables are the chosen solvent (ethanol, butanol, acetone, ethyl acetate, and water, among others), temperature, pressure, and extraction time. In addition to the extraction of BACs, this technique can also allow the hydrolysis of carbohydrate polymers, which can be used further [73].

*Supercritical fluid extraction (SFE)*: The principle of this extraction technique is that a fluid above the critical conditions behaves like a gas and a liquid. A supercritical fluid has a high diffusion coefficient, and low viscosity and surface tension. This leads to increased penetration into the sample matrix and favorable mass transfer. Carbon dioxide has been studied as an ideal solvent for SFE, given the low critical conditions (approximately 30°C and 70 bars), which allows operating at moderate pressures, and having an easy and favorable regeneration of the solvent. Due to its polarity, it is used for the extraction of lipids, fats, and non-polar substances; however, the use of a co-solvent has been done in order to increase the span of compounds that can be extracted [74]. The most important variables are temperature, pressure, particle size, moisture content, extraction time, flow rate, and solvent-to-feed-ratio [75].

*Ionic liquids (IL)*: More than a specific technique for extraction, ionic liquids are solvents that are being studied to replace conventional solvents. These are an organic cation paired with an organic/inorganic anion (organic salt), in a liquid state [76]. They have properties such as poor conduction of electricity, non-polarity, high viscosity, low vapor pressure, low combustibility, high thermal stability, and good solvating properties for polar/non-polar compounds. These compounds are also known as "designer solvents", given that they can be customized for specific extractions of target compounds based on the unique chemical functional groups [77].

### 3.3.2. Production Techniques

This section describes the technologies for the production of BACs, specifically antibiotics, which are metabolized by a microorganism and then extracted and purified. Regarding the production technologies of antibiotics, the three methods are natural fermentation, semi-synthetic, and synthetic production.

Antibiotics produced via *natural fermentation* are often obtained at scales around 100,000–150,000 liters or more in specific growth mediums. The fermentation conditions need to be managed carefully in order to control the population growth of the microorganism and hence obtaining high yields of secondary metabolites. Then, the antibiotic must be extracted and purified, which is dependent on the solubility of the compound in organic solvents; if not, other processes such as ion exchange, adsorption, or precipitation must be performed. Aminoglycosides are an example of this antibiotics group [78].

*Semisynthetic* antibiotics are modifications of natural compounds and they represent most of the antibacterials currently being produced. The modifications are aimed to maximize the efficacy of the drug, the production yield, and the potency of the antibiotic. These modifications can be the genetic modification of microorganisms, and further chemical modifications of the compounds produced in the fermentation, among others. Beta-lactam antibiotics are an example of these antibiotics, which include penicillins (produced by *Penicillium* fungi), cephalosporins, and carbapenems [79]. *Synthetic* antibiotics are produced only by chemical synthesis. Sulfonamides, quinolones, and oxazolidinones are some examples of this antibiotics group [80].

## 4. Sustainable Production: Feedstock Composition and Further Valorization

As it was observed in the previous section, there is a wide variety of sources for the extraction/production of BACs. However, in order to avoid residue production and to make the process sustainable, it is necessary to consider an integral use of the different components of the raw material, after the respective extraction/production of the BACs. For this, it is necessary to describe the main components of the feedstocks.

For the extracted BACs, as they come in a natural matrix formed by many other components, the focus is on the identification of these components and their possible uses. However, the approach is different for the production of BACs (antibiotics). In this case, as it is generally a fermentative-based process, the substrate (carbon source) and nutrients play one of the most important roles. Therefore, aiming for the sustainability of the process, there are many possible sources of carbohydrates that could be used as substrate. Second-generation raw materials could be a source of substrates (hexoses, pentoses, glycerol, starch, etc.). This becomes a point of coincidence between the two approaches to obtain BACs, as plant-based feedstocks can offer the possibility of extracting BACs and the remaining fractions can be used to produce fermentable sugars that can be the substrate to produce antibiotics.

### 4.1. Feedstock Composition

The components of the matrix of the feedstock becomes of great importance, because a given raw material has a specific composition, which therefore, will determine the possible platforms and products that could be obtained. Figure 2 shows different feedstock generations, building blocks (platforms), and possible products that can be obtained based on the composition of biomass. For biomass, the main components of a feedstock are lignocellulose (cellulose, hemicellulose, and lignin), starch, oils, and proteins.

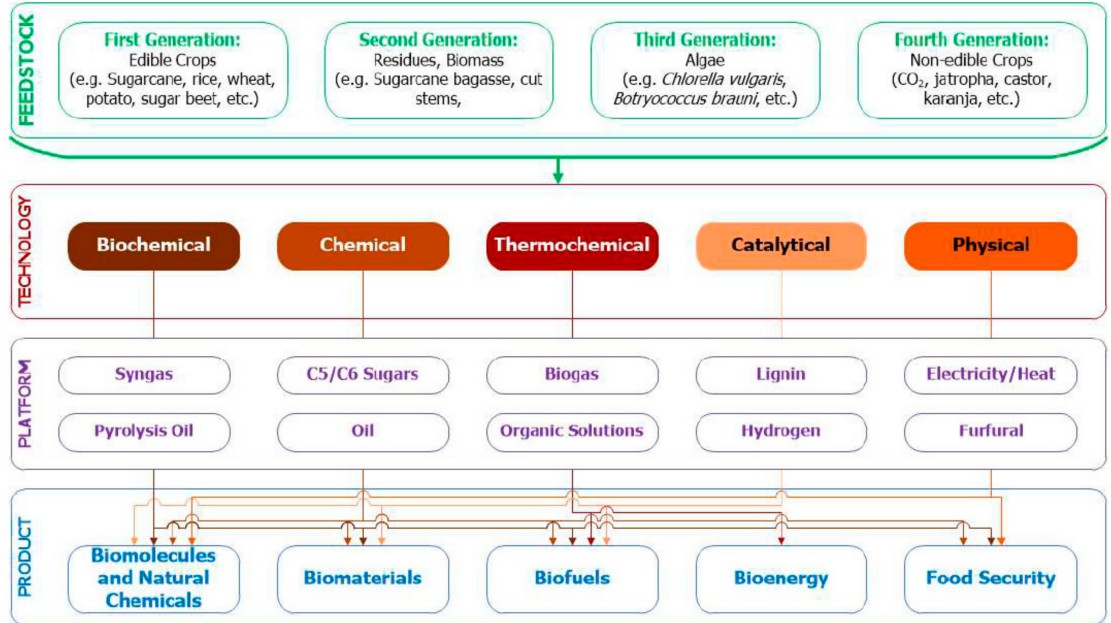

**Figure 2.** Biomass generations, building blocks (platforms), and possible products that can be obtained based on the composition of biomass. Taken from (Cardona et al., 2020) [7].

Mainly three polymers build up the lignocellulosic complex: Two carbohydrate polymers (cellulose, hemicellulose) and an aromatic polymer (lignin). Cellulose and hemicellulose are polysaccharides composed of hexoses and pentoses that can be used as carbon source [81,82]. Agricultural feedstocks are mainly formed by starch and glycan, while their wastes and forestry materials are formed commonly with more lignin.

Cellulose is a homo-polymer of D-glucose subunits, linked by β-1,4 glycosidic bonds. The structure consists of a crystalline and an amorphous fraction. Hemicellulose is a hetero-polymer of pentoses (xylose and arabinose) and hexoses (mannose, glucose, and galactose); the most dominant component is xylan. Lignin is a polyphenolic structure whose role is providing structural strength in plants. Cellulose correspond to the fiber, linear structure and hemicellulose is the binding material between both polymers [73]. Regarding the possible applications, hemicellulose is generally hydrolyzed into sugars. Hemicellulose is the most thermal-chemically sensitive fraction, while lignin and cellulose require harsher or more specific pretreatments to be hydrolyzed. Base-catalyzed hydrolysis, Organosolv, sulfite, and Kraft are some of the pretreatments used to solubilize lignin, whereas enzymatic hydrolysis and acid hydrolysis are applied to solubilize cellulose into glucose.

*Starch* is a glucose polymer joined by glycosidic bonds. Most plants produce this polysaccharide as an energy storage. It is made up by a linear fraction (amylose) and a branched fraction (amylopectin). The most used application for this compound is as food, but lower quality starches are also used for further hydrolysis and production of fermentable sugars, for adhesives, and as additive in textile industry [83].

### 4.2. Further Valorization

In addition to the main structural components of plant feedstocks mentioned in the previous section, other components contained in some parts of the plant may have applications. To mention some examples, seeds that can be found in raw materials such as grapes, hemp, olives, and pumpkin, have a high content of oils, which can be further extracted; these oils can have food applications and for the production of biodiesel [84,85]. Another example can be pectin, which can be found in apple pomace, citrus peels, and blackberry; it is a polysaccharide recognized as a food additive, used as a gelling, stabilizing, or emulsifying agent in multiple industrial processes [86–88].

Another source of possible valorization byproducts can come from the fermentation stage of BACs production. In this case, it was shown before that the fermentation volume is around 150,000 liters, which implicates the growth of a significant amount of microorganism/biomass. Despite the fermentation products are extracellular (expulsed to the medium) or intracellular (maintained within the membrane and hence requiring a further extraction stage), when the fermentation finishes, the biomass represents a significant amount of solids that must be separated. This cake could have food applications as probiotics [89] and thermochemical applications [90]. When grains and cereals are used as feedstock for substrate production, after the respective fermentation, the solid cake has a high content of protein and it can be used for feed purposes [91].

### 4.3. Final Valorization: Energy Production and Nutrient Recovery

Finally, after the subsequent valorization of different compounds and fractions of the feedstock, the remaining solids that cannot be further separated still represent a source of elemental components (carbon, nitrogen, hydrogen). These can still be used under thermochemical and biochemical processes focused on the production of energy/power and heat. Recovering energy from this residual biomass provides a principal strategy to cover a share of the energy requirement of the process, mitigating the global environmental problems [92]. Pyrolysis, gasification, and anaerobic digestion are some of the processes that can be applied.

*Pyrolysis* consists of heating a given material above the decomposition temperature (up to 500 °C), breaking the chemical bonds. It produces solids (char), condensable liquids (tar), and gasses. Pyrolysis oil can be used for biofuel applications and the produced syngas can be further used for Fischer–Tropsch reactions or fermentation [93]. *Gasification* consists of converting the organic material into carbon monoxide, hydrogen, and carbon dioxide by heating to temperatures above 700 °C without combustion, with a controlled amount of oxygen and/or steam. The product is mainly syngas at high pressure and temperature, which can be used to produce energy, and further used in Fischer–Tropsch reactions or to purify the hydrogen [94]. Finally, *anaerobic digestion* is a biological process through which

microorganisms break biodegradable material. One of the products is biogas, which can be used to produce electricity and heat, or purified to natural gas and transportation fuels. In addition, the solid sludge can be further treated to be used as fertilizer [95].

The use of these remaining solids as fertilizer has a significant potential for the recovery of essential macro elements as nitrogen, phosphorus and potassium, and trace elements and minerals as S, Cl, Na, Ca, Mg. The sludge resulting from anaerobic digestion and the ashes resulting from the thermal processes contain these trace elements, although some volatile components may be lost under processes operating at higher temperatures [96]. This is a very important stage to be considered in the use of biomass, as these minerals are derived from finite resources; hence, their recovery and reuse became of great importance to sustain future food security and long-term availability [97,98].

### 4.4. Examples of Feedstock Valorization According to Different Composition Fractions

After describing the different components, Table 3 shows examples of different valorization schemes and uses proposed for different additional fractions of raw materials that are used to obtain BACs. It also describes the applied technologies and the aimed platforms/products. It is important to mention that most of the referenced studies correspond to standalone processes of a single application to the proposed residue. Although, in some cases, the proposed uses correspond to cascade-use biorefinery schemes. However, this broad number of applications of the different fractions is an excellent starting point to propose biorefineries where each valuable fraction is used and exploited, until closing the balance with energy-driven stages to supply energy for the process.

**Table 3.** Valorization of different additional fractions of feedstocks used to obtain bioactive compounds.

| Source | Applied Technology | Fraction | Platforms/Products | Ref. |
|---|---|---|---|---|
| Industrial hemp—threshing residues | Pressing | Seeds | Oils | [99] |
| | Enzymatic hydrolysis | Lignocellulosics | Glucose (substrate) | [41] |
| | Acid hydrolysis | Lignocellulosics | Ethanol Succinic acid | [100] |
| Wheat straw | Organosolv | Lignin | Nano-lignin | [31] |
| | Liquid hot water | Hemicellulose | C5-sugars (substrate) | [101] |
| | Enzymatic hydrolysis | Cellulose | Glucose (substrate) | [102] |
| | Anaerobic digestion | Solids | Heat and power; Methane | [103] |
| Coffee residues | Dehydration | Hemicellulose | Furan-based products Furfural | [104] [105] |
| Rice straw | Acid hydrolysis Enzymatic hydrolysis | Lignocellulosics | Xylitol | [106] |
| | Liquid hot water Dilute-acid hydrolysis | Hemicellulose | C5-sugars (substrate) | [107] |
| Grape vine shoots | Liquid hot water | Cellulose | Ethanol | [108] |
| | Enzymatic hydrolysis | Hemicellulose | | |
| | Alkali hydrolysis | Lignin | Lignin | |
| | Anaerobic digestion | Solids | Heat and power Methane | [109] |
| | Gasification | Solids | Syngas | [110] |
| Grape pomace | Anaerobic digestion | Solids | Heat and power Methane | [111] |
| Apple pomace | Enzymatic hydrolysis | Pectin | Pectin | [112] |
| | Enzymatic hydrolysis | Cellulose | Glucose (substrate) | |
| | Anaerobic digestion | Solids | Heat and power Methane | [113] |
| Blackberry pulp | Acid hydrolysis | Hemicellulose | Xylitol | [114] |
| | Enzymatic hydrolysis | Cellulose | Ethanol | |
| Orange residues | Enzymatic hydrolysis | Pectin | Food | [115] |
| | Acid hydrolysis Enzymatic hydrolysis | Cellulose Hemicellulose | Succinic acid | [116] |
| | Pyrolysis | Solids | Pyrolysis oil Char | [117] |
| Olive stone | Acid hydrolysis | Hemicellulose | Xylitol Furfural | [113] |
| | Enzymatic hydrolysis | Cellulose | Ethanol PHB | |
| | Anaerobic digestion | Solids | Heat and power Methane | |
| Fermentation by-products (DDGS) | Thermal-mechanical | Protein and fiber | Feed | [91] |

DDGS: Distiller's dried grains with solubles; PHB: Polyhydroxybutyrate.

### 5. Study Case: Selection of a Feedstock for a Sustainable Biorefinery for the Production of Bioactive Compounds

The next step consists of coupling the different elements analyzed in the previous sections (BACs, sources, and technologies) to select possible feedstocks for sustainable biorefineries to produce BACs. This section will show three examples of feedstocks that can be used for BAC-producing biorefineries and some possible valorization schemes that can be proposed. This selection analysis will be performed based on the conditions and crops available in Austria. Since 2015, the European Commission adopted an action plan to accelerate the transition towards a circular economy, and encourage sustainable economic growth, and biomass and bio-based materials is one of the five priority sectors of this strategy [118]. Within this frame, the Austrian government has promoted well-established strategies focused on recycling, waste-to-energy facilities for energy recovery, and circular economy. In addition, it is one of the countries with higher intensive use of land and, despite not having the largest agricultural land area, still produces enough to cover the internal demand and generating surplus in crops as wheat [119,120]. This context indicates that Austria has an interesting outlook for future research and investment in topics that promote the transition into a bio-based economy, through a sustainable production of value-added products.

*5.1. Raw Material Selection*

Based on the list of raw materials described in Tables 1 and 3, the next step consists of searching the production in Austria. This information is presented in Figure 3. The selection of the raw material should be done in terms of different criteria: Availability, present BACs, and possibility of further valorization, among others.

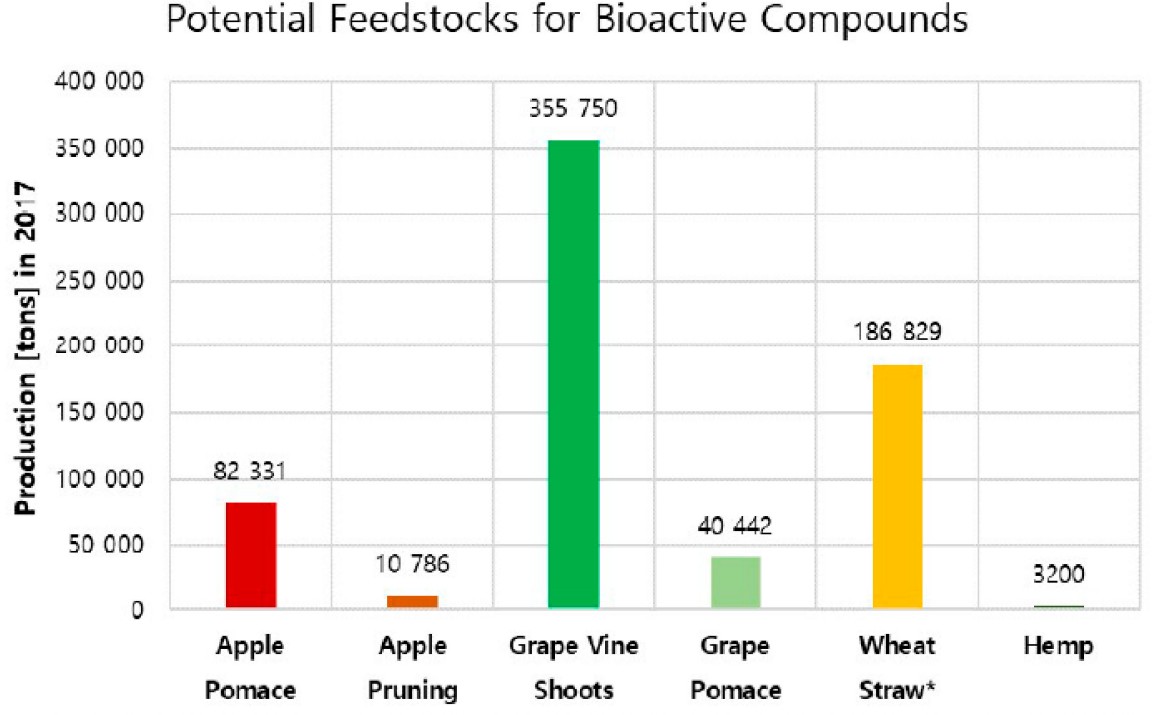

**Figure 3.** Production of different potential feedstocks to obtain bioactive compounds in Austria. Data collected from FAOSTAT [121] and EIHA (European Industrial Hemp Association) [122]. * For scale purposes, the production of wheat straw was divided by a factor of 10.

Table 1 shows the BACs present in each of the raw materials shown in Figure 3. In terms of the *availability*, wheat straw (WS) has the highest production (1.8 million tonnes), which establishes it as an excellent feedstock to obtain nano-lignin. Wheat is the most grown food crop around the world, with

approximately 218.5 million ha and an overall production of around 770 million tonnes in 2017. It is considered as one of the most important sources of carbohydrates and vegetal protein in human consumption [121]. For the same year, Austria produced around 1.4 million tonnes of wheat in an area of around 295,000 hectares. This is approximately 1.8 million tonnes of wheat straw generated as residue. In addition, WS has significant contents of cellulose and hemicellulose, which represent approximately 63 wt% of the raw material [123]. This represents a significant potential to obtain a sugar platform that could be further used as substrate for a microorganism fermentation to produce antibiotics, and the remaining solids could be further used for energy production.

In terms of the *present bioactive compounds*, hemp (HE) represents the most interesting option, as currently cannabinoids—especially cannabidiol (CBD)—have gained interest in the pharmaceutical industry, which makes it a specialty chemical with high added value that could be an economic driver in a biorefinery [41]. In addition, CBD is contained mainly in the leaves/blossom, which leaves the stem part available to produce fiber and materials [124], and after the respective extraction of the CBD, the remaining solids can be used to obtain sugars and follow the fermentation strategy also described for WS.

In terms of the *possibility of further valorization*, the remaining raw materials offer different options, pros and cons; but, the grape vine shoots and the apple pomace highlight due to their higher production compared to the apple pruning residues and the grape pomace. Vine shoots consist of the stem and leaves that are cut after the grapes are harvested. This offers a lignocellulosic platform in the stem that can be used for fiber, lignin, and sugar production. The remaining solids after the extraction of BACs in the leaves could be also used further for sugar production. Apple pomace consists of the wet residue resulting from the processing of apples for juices and jam industry. It contains carbohydrates and pectin, which could be used as substrate and purified, respectively. In terms of the present BACs, as shown in Table 1, both groups of residues (coming from grapes and from apple) have compounds of high interest, for example, stilbenes such as resveratrol, and dihydrochalcones such as phloridzin, respectively. Therefore, grape vine shoots could be considered a better option as it offers more platforms to be used.

## 5.2. Proposed Biorefineries

After the selection of raw materials, the next considered step is proposing schematic theoretical biorefineries, focused on the identification of the platforms that can be obtained from these raw materials, the fractions that can be used to obtain BACs, and the valorization of the remaining fractions. Figure 4 shows the general scheme of a biorefinery for the production of BACs under this scheme. It is important to mention that the specific selection of technologies for each step cannot be generalized to a single set of technologies. This decision must obey a more detailed sustainability assessment of the different possible configurations of the technologies described along Section 3.3. However, based on the current possible uses reported in Table 2 for each fraction, some example technologies are mentioned. Figure 5, Figure 6, and Figure 7 show the general scheme of a biorefinery for the production of BACs using wheat straw, hemp, and grape vine shoots as feedstock, respectively.

For the case of wheat straw (Figure 5), the initial step is the extraction of lignin and further valorization to produce nano-lignin. Then, the remaining solid can be used in two ways: First, to obtain C5- and C6-sugars for the production of antibiotics; second for fiber valorization. Then, after one of these options, the remaining solids can be further used to produce energy. This feedstock offers a comparative benefit with respect to the hemp and the vine shoots, which is the significant amount produced. This is an important element, especially considering the energy-production stage, as the scale is a factor that strongly influences the techno-economic performance of technologies/processes, as well as other factors as the local availability, the size of the biorefinery plant, and centralized/decentralized processing schemes.

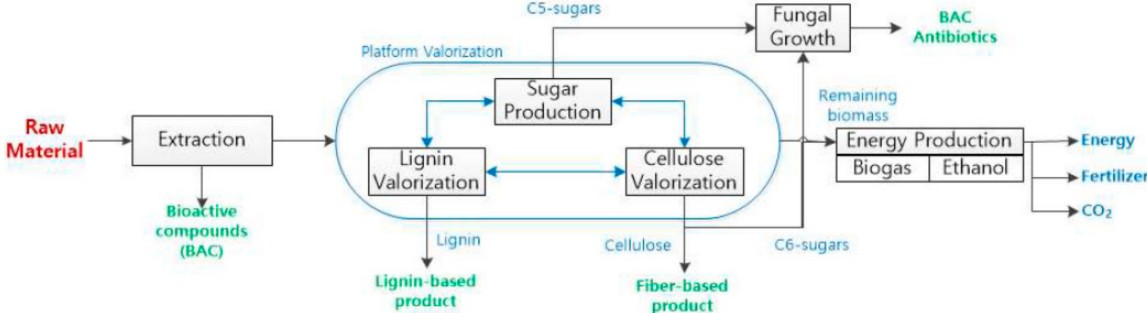

**Figure 4.** General scheme of a biorefinery for the production of bioactive compounds. BAC: Bioactive compounds.

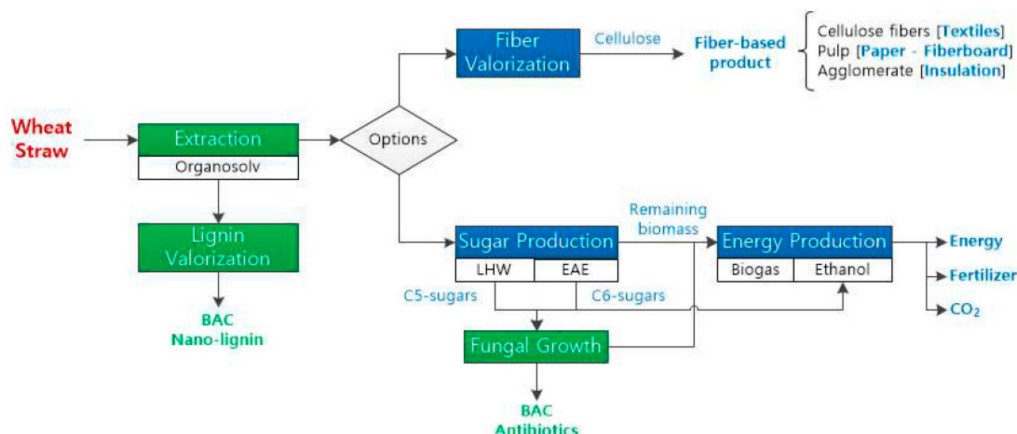

**Figure 5.** General scheme of a biorefinery for the production of bioactive compounds using wheat straw as feedstock. BACs: Bioactive compounds; EAE: Enzymatic-assisted extraction; LHW: Liquid hot water.

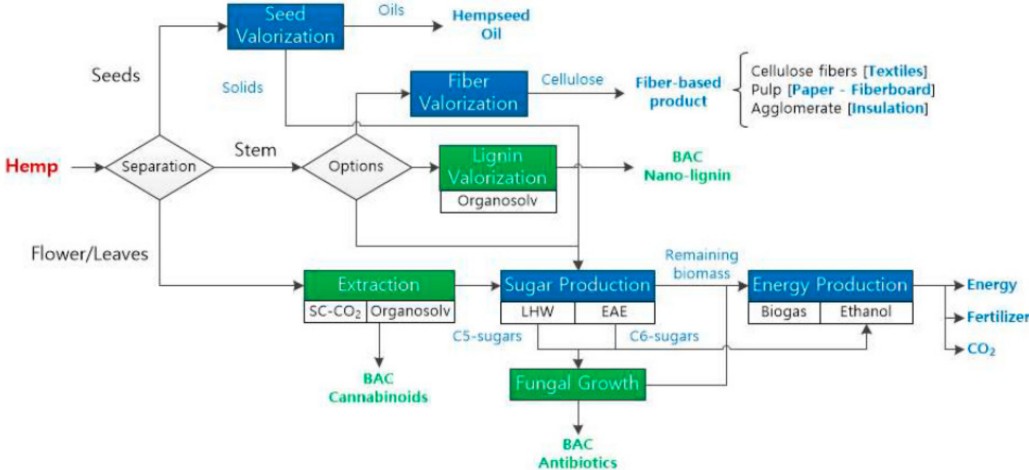

**Figure 6.** General scheme of a biorefinery for the production of bioactive compounds using hemp as feedstock. BACs: Bioactive compounds; EAE: Enzymatic-assisted extraction; LHW: Liquid hot water; SC-CO$_2$: Supercritical CO$_2$.

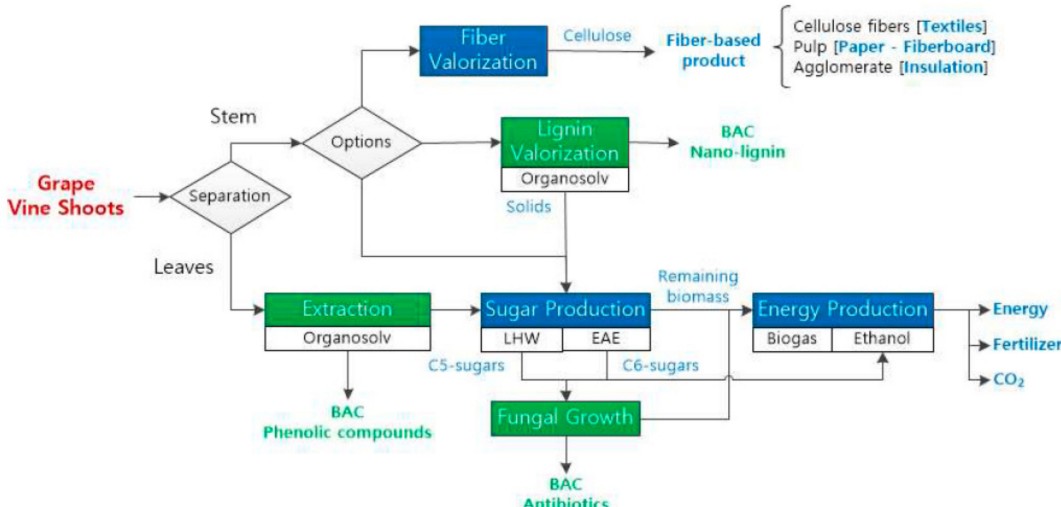

**Figure 7.** General scheme of a biorefinery for the production of bioactive compounds using grape vine shoots as feedstock. BACs: Bioactive compounds; EAE: Enzymatic-assisted extraction; LHW: Liquid hot water.

For the case of hemp (Figure 6), an initial separation of the plant sections (stems, flower/leaves, and seeds) is proposed. The first fraction that can be valorized is the seeds, which can be pressed to obtain the oil and used for food applications. From the stem, based on the composition and lignin content, an extraction of the lignin can be done in order to produce nano-lignin, and then a fiber valorization of the cellulose, in order to obtain products for material applications (textiles, pulp, and agglomerates). A third application can be the treatment to obtain C5- and C6-sugars to be used as substrate. From the flower/leaves mixture, it is necessary to extract the cannabinoids; Organosolv extraction and supercritical $CO_2$ are technologies that can serve this purpose. Then, the remaining solids, coupled with the solid remnants from the oil extraction of the seeds, can be used for sugar production that can be the substrate for a fungal growth aimed to produce antibiotics. Finally, the remaining biomass can be taken to an energy-production stage (e.g., biogas) or even considered for a hydrolysis for the production of ethanol.

For the case of the grape vine shoots (Figure 7), the strategy can be similar to that proposed for hemp, as this feedstock is formed also by two sections (stem and leaves). Therefore, the initial step is a separation of the sections of the biomass. From the stem, the options can be an extraction of the lignin, followed by the further treatment to obtain C5- and C6-sugars or a fiber valorization. From the leaves, the phenolic compounds are extracted. Then, the remaining solids can be used to obtain sugars for the production of antibiotics. Finally, the remaining biomass can be taken to an energy-production stage. In this case, it is still necessary to identify a target bioactive compound as different reported bioactive compounds that can be obtained—resveratrol, Kaempferol, quercetin—have interesting applications.

## 6. Conclusions and Outlook

It was clearly shown that there is a wide variety of feedstocks offering the possibility to obtain bioactive compounds, especially with a broader panorama of what is considered as bioactive compound. The inclusion antibiotics—produced BACs—opens new connections for the use of substrates obtained from second-generation feedstocks. In addition, considering the application of lignin as a bioactive compound opens the market for this fraction of the feedstock and may increase the added value, compared to other applications that it currently has.

Regarding the processes, it is necessary to start proposing cascade, biorefinery schemes that allow valorizing the different fractions. The selection of specific technologies for the different stages of a biorefinery cannot be generalized to a single set of technologies. This decision must obey a sustainability

assessment of different configurations of the technologies. As there are multiple possible configurations and setups of technologies, the key indicator since the design stage should be the assessment of the sustainability, in terms of the technical, economic, environmental, and social performance.

Agroindustrial residues highlight as feedstocks having various platforms that can be valorized to obtain different types of products. This is a very interesting outlook for biorefineries, given that a portfolio of products confers adaptability to a process and decreases the dependence on a single product. Bioactive compounds may become economic drivers that allow decreasing the process scale and even considering small-scale decentralized biorefineries close to the location where the feedstock is produced. This would decrease the environmental burden associated to the transport of feedstocks. However, a deeper supply-chain and logistic analysis must be coupled to the design and the analysis of the life cycle of the biorefinery in order to determine the feasibility in this aspect.

In terms of the products for the biorefineries, it is necessary to evaluate simultaneously the quality and applicability. To mention some examples, pharmaceutical and nutritional applications need to reach further evaluation stages to be considered valid for disease treatments or as nutritional supplements. Regarding fiber-based applications, further research in terms of the quality of the obtained fibers is required, as this is a determinant factor for the application field.

Finally, as this biorefinery scheme for the production of bioactive compounds offers the possibility of obtaining a platform of sugars to be used as substrate, this could be an excellent option to expand the palette of products that can be obtained. This production could cover not only antibiotics, but also other fermentative-based products for example, fine/specialty chemicals such as erythritol, biofuels such as ethanol and butanol, or materials such as polyhydroxyalkanoates.

**Author Contributions:** S.S.-L. wrote and edited the manuscript with significant input and editing from A.M., M.M., and and A.F.

**Funding:** This research received no external funding.

**Acknowledgments:** The authors acknowledge Vienna University of Technology (TU Wien) for the funding of the Doctoral College "Bioactive", under which this research was performed and TU Wien Bibliothek for financial support through its Open Access Funding Programme. Open Access Funding by TU Wien.

**Conflicts of Interest:** The authors declare no conflict of interest.

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
