# Peer review of "A Review on the Feedstocks for the Sustainable Production of Bioactive Compounds in Biorefineries"

_sustainability, doi:10.3390/su11236765_

Round 1
Reviewer 1 Report
Dear Authors,
In general, the paper is well written and structured. It contains the main introductory and theoretical information regarding biorefineries and bioactive compounds. I would like to make some comments and propose some modifications.
- Language: few sentences require another revision, such as in lines 27 (bio-based systems?) and 62 (Bac can be obtained from many…)
- lines 120 and 132: Please state to each section the “Section 0” is linked to.
- line 155: The examples of synthetic molecules include coumarin, which is natural compound. Please revise this sentence.
- lines 299-305: Although the discussion about the role of solvent composition in the extraction of bioactive compounds is relevant must be highlighted, other relevant variables must be considered (such as temperature, solid/solvent ratio, time, and number of extractions). Please add more information regarding these variables. In addition, update the reference of this paragraph
- lines 335-339: Please a short description of the mechanism of pore formation by exposing a matrix to PEF
- section 4: This section is the core of the manuscript and deserve more attention. It is important to improve the information discussed in this section in order to present more than definitions of compounds and processes. It is important to indicate and discuss the most promising sources and bioactive compounds from the studies available in literature. Additionally, it is important to show indicators such as proportion of bioactive compound in the matrix, extraction yield, energy consumption for both conventional and green strategies… Please revise this section.
- Table 3: The information is interesting from a general point of view but does not support the discussion in this section. Several products (methane, syngas, xylitol, pyrolysis oil…) could be considered as of minor importance in relation to bioactive compounds (main focus of the review) and their importance in the manuscript should be justified. Moreover, the table does not contain any information about the use of green technologies (SC-CO2, ultrasound…) or green solvents. Some of the process were not described in the previous sections of the manuscript (acid hydrolysis, for instance) Please reformulated this table and make appropriate changes in the discussion.
- section 5: Another core section of the manuscript like section 4. Please improve this section and discuss the findings of other studies regarding the proposed or related biorefinery (Figures 5-7).
- lines 505-508: Why Austria? Please justify its importance for the manuscript.
- figures 4-7: Please indicate the meaning of each abbreviations used in these figures
- conclusion: Its currently form is too general and does not reflect the meaningful findings reported in recent studies. Please revise this section of the manuscript.
Author Response
Ref. No.: Sustainability-638436.
Title: A review on the feedstocks for the sustainable production of bioactive compounds in biorefineries.
SUSTAINABILITY.
Dear Prof. Dr. Marc A. Rosen
Editor-in-Chief
We appreciate the time that the reviewers have dedicated to evaluate the paper since their contributions are valuable to improve the scientific quality of our work.
According to the reviewers' comments, we have done the following changes to the paper, as can be seen in the attached document.

Reviewer 2 Report
This manuscript reviews the use and potential of different feedstocks for the production of bioactive compounds (BAC) within a biorefinery perspective. In general, it is well written and discussed, and represents an interesting contribution for the research in its field. Furthermore, it highlights different aspect for the production of BACs that are directly related to the feedstock such as the BAC type or the conversion technology to be used. Notwithstanding, the following minor comments should be considered prior to publication:
- Affiliation 2 is not listed in the title page and should be double checked.
- Keywords are intended to gain visibility of the work. However, present keywords are already mentioned in the title and should be therefore replaced. The following suggestions can be considered: Bio-based industry, biomass, industrial and urban residues, etc.
- Abreviations should be defined first time they appear in the text and be used subsequently. This applies for instance to “bioactive compounds: BAC”.
- Lines 119 and 132: references to Section 0 are confusing and should be double checked.
- Line 407: Hexoses, pentoses, glycerol, and starch are not second-generation substrates, but sugars that can be obtained from second-generation substrates (as explained in the subsequent sentence, lines 409-410). This should be reformulated.
Author Response

(The authors gave the same response as above.)

Round 2
Reviewer 1 Report
Dear Authors,
The suggested modifications were incorporated in the new version of the manuscript. In my point of view, the manuscript is acceptable.